# Phenotypes and Endotypes in Sarcoidosis: Unraveling Prognosis and Disease Course

**DOI:** 10.3390/biomedicines13020287

**Published:** 2025-01-24

**Authors:** Ilias C. Papanikolaou, Konstantinos Chytopoulos, Dimitrios Kaitatzis, Nikolaos Kostakis, Anastasios Bogiatzis, Paschalis Steiropoulos, Fotios Drakopanagiotakis

**Affiliations:** 1Department of Pneumonology, Kerkyra General Hospital, 49100 Corfu, Greece; icpapanikolaou@hotmail.com; 2Department of Pneumonology, Medical School, Democritus University of Thrace, University General Hospital Dragana, 68100 Alexandroupolis, Greecesteiropoulos@yahoo.com (P.S.)

**Keywords:** sarcoidosis, phenotypes, endotypes

## Abstract

Sarcoidosis is a multi-system granulomatous disease of unknown etiology. In genetically susceptible individuals, the precipitating factors generate, via immunity mechanisms, a host granulomatous response. The granuloma, for unknown reasons thus far, may resolve or may persist and lead to organ damage and fibrosis. Infectious agents, occupational exposure, obesity, smoking and genetic factors are implicated in the pathogenesis of sarcoidosis. Macrophages are important in granuloma formation, and their M1/M2 phenotype is associated with the prognosis of the disease. CD4^+^ T helper cells play a central role in the pathogenesis of sarcoidosis. The major contributors appear to be Th1 and Th17.1 cells, whose microenvironmental behavior is dictated by the secretions of macrophages and dendritic cells. Higher levels of Th1 and Th17.1 cells are associated with chronic disease and resistance to corticosteroid treatment. In recent years, advances in the phenotyping of sarcoidosis with the help of HRCT, PET-CT and lung function tests have provided us with a better understanding of the disease. Genetic phenotyping performed by the GenPhenReSa consortium and the SAGA study has led to the recognition of new, distinct phenotypes. The reconstitution of dysregulated autophagy through persistent m-TORC-1 pathways may be a new treatment target in sarcoidosis.

## 1. Introduction

Sarcoidosis is a multi-system granulomatous disease of unknown etiology [1]. In genetically susceptible individuals, the precipitating factors generate via immunity mechanisms a host granulomatous response. This granuloma, for unknown reasons thus far, may resolve or may persist and lead to organ damage and fibrosis [1]. The mechanisms determining the natural course of the granuloma are not exactly known. In recent years, advances in the pathophysiology of sarcoidosis and new pathogenetic mechanisms have come to light. This knowledge is followed by the more sophisticated phenotyping and genome-associated endotyping of sarcoidosis [1,2,3]. These developments and their future permutations are the focus of this review.

The diagnosis of sarcoidosis requires compatible clinical and radiological features, tissue pathology confirming non-caseating granulomatous inflammation in the absence of infection and foreign bodies and the exclusion of other diagnoses with similar presentation. Compatible presentation refers mainly to pulmonary involvement, as the lungs are insulted in 95% of cases, with non-specific symptoms. Chest radiography and computed tomography typically reveal nodular opacities of the upper- and mid-lung fields bilaterally, often confluent, in a peri-bronchovascular, centrilobular or subpleural distribution, along with mediastinal and hilar lymphadenopathy. Other typical clinical presentations include lupus pernio, Löfgren syndrome and Heerfordt syndrome. Other sites or organs commonly involved are the extra-pulmonary lymph nodes, liver, eyes, skin, heart, kidneys and the central nervous system [4].

## 2. Etiology

The exact etiology of sarcoidosis is still unknown, but it is thought to be a multifactorial disease due to the combined effect of genetic predisposition, environmental factors and immune system dysfunction. Sarcoidosis may be the result of an immunological cascade due to exposure to an unknown antigen in genetically vulnerable individuals. Such antigens may include metals such as silica and infectious agents, such as Propionibacterium species and Mycobacterium [1,2]. 

### 2.1. Genetic Background

The genetic background seems to play a decisive role in the development of sarcoidosis, as the estimated risk is 5-fold higher among siblings [3] and 7-fold higher for dizygotic and 80-fold higher for monozygotic twins [5]. Additionally, numerous studies have linked various human leukocyte antigen (HLA) variants to increased risk of sarcoidosis, as well as to the specific organs affected and the severity of the disease. Characteristically, *HLA-DRB1*03*, **0301* or **1501* are associated with Löfgren syndrome [6,7,8,9,10]. Similarly, *HLA-DRB1*07*, **14*, **15*, **01* and **03* and *DQB1*0602* are linked with higher likelihood of progressive pulmonary sarcoidosis [6,11].

Tumor necrosis factor (TNF)-α and interleukin (IL)-23, which drive the Th1 and Th17 immune responses, respectively, are implicated in granulomatous diseases. TNF-α polymorphisms (*308AA* and *rs1800629*) are associated with both increased risk of sarcoidosis [12,13,14,15] and pulmonary disease progression [13], while a IL-23 polymorphism (*rs12069782*) increases sarcoidosis risk [16] but appears to protect against pulmonary disease progression [17]. The chemokine receptors CCR2 and CCR5 single-nucleotide polymorphisms (SNPs) show links to Löfgren syndrome [18] and to the severity of pulmonary sarcoidosis [7]. Polymorphisms in Toll-like receptors (TLRs) also correlate with sarcoidosis risk [19,20,21]. Other genes, including *Butyrophilin-like-2* (which influences T-cell regulation), *SLC11A1* (which encodes macrophage membrane protein) and *Annexin A11* (which is involved in cell apoptosis), are all associated with different sarcoidosis phenotypes [22,23,24]. Vimentin, a structural protein encoded by the *VIM* gene, may be involved in the pathogenesis of sarcoidosis. Tertiary lymphoid structures enriched in vimentin are present in sarcoid lung tissue and, at the same time, are absent in healthy lungs. In addition, sarcoid lung is linked to the presence of anti-vimentin antibodies and clonal lymphocytes with specific characteristics [25]. Future studies with larger samples of patients and comprehensive genetic sequencing, such as whole-exome sequencing, are needed to deepen the understanding of sarcoidosis genetics. 

### 2.2. Pathogenic Agents

The multi-systemic nature of sarcoidosis, along with its familial and geographical clustering and its clinical and histological resemblance to mycobacterial infections, has led to the hypothesis that an infectious agent may be responsible for the disease. An aerosolized microbial agent is a compelling theory, as it could explain the environmental, geographic and seasonal variations, as well as the systemic symptoms of sarcoidosis [2]. Mycobacterial species and Propionibacterium acnes are considered the main infectious agents associated with the pathogenesis of sarcoidosis. Many studies using molecular techniques have found the presence of mycobacterial RNA or DNA in sarcoid granulomas, even though mycobacteria have never been cultured or microscopically found in sarcoid granulomas [26,27]. Research has also identified specific anti-mycobacterial immune responses in sarcoidosis patients but not in controls, and in vitro studies have shown similar immune profiles when cells from patients were exposed to mycobacterial agents [26,28,29]. In contrast to mycobacteria, Propionibacterium acnes has been successfully cultured from sarcoid tissue. However, P. acnes is a common commensal organism found both in healthy individuals and in other granulomatous diseases [30,31]. Other studies have also detected, in sarcoid tissues, antigens from various bacteria and rarely from fungi and viruses [2]. To explore this hypothesis, a randomized, double-blind, placebo-controlled trial was conducted, testing the effectiveness of a combination of antimicrobial treatment against mycobacteria (levofloxacin, ethambutol, azithromycin and rifampin) in 97 patients with active pulmonary sarcoidosis. The study showed no statistically significant differences in the change in FVC among the 49 patients receiving treatment compared with the 48 receiving the placebo [32]. The lack of efficacy in antimicrobial therapy has led researchers to propose that persistent infection may not be essential for sarcoidosis to develop. Instead, initial exposure to an infectious antigen might trigger an immune cascade that sustains the disease, explaining the reason why immunosuppressive treatments are generally effective, whereas antimicrobial therapies are not [2]. 

### 2.3. Occupational Exposure

Several studies, including epidemiological research, case reports and immunological investigations, have implicated a correlation between work-related exposure and sarcoidosis. One of the earliest large-scale investigations, the ACCESS study in the U.S., which was a retrospective, multicenter case–control study involving 720 sarcoidosis patients, has revealed correlations between sarcoidosis and exposure to insecticides, dust, mold and livestock [33]. Subsequent studies have reported a higher risk among firefighters, agricultural, metal and construction workers [34]. A remarkable increased risk of sarcoidosis among firefighters at the World Trade Center (WTC) on September 11 2001 was reported in a cohort study using historical controls. Specifically, at the Fire Department of New York, the annual incidence of sarcoidosis increased from 15 individuals per 100.000 in the spacetime of 15 years prior to 2001 to 85 individuals per 100.000 in the year after 2001 [35]. Furthermore, silica and metal dust are strongly associated with certain cases of sarcoidosis. Many case–control studies have reported higher risk of exposure to silica in patients with sarcoidosis compared with controls [36,37]. 

### 2.4. Lifestyle Factors

Among the various lifestyle factors linked to sarcoidosis, the strongest evidence points to obesity and smoking. Obesity, a complex systemic condition, involves immunological and hormonal changes that contribute to chronic inflammation, insulin resistance and shifts in immune function. Conditions like asthma, rheumatoid arthritis and lupus have been associated with obesity [38,39,40]. However, establishing a direct causal link between obesity and sarcoidosis has been challenging. Symptoms commonly seen in obesity, such as exercise intolerance, fatigue, poor sleep and shortness of breath, overlap significantly with those of sarcoidosis. Furthermore, sarcoidosis patients often experience chronic fatigue and reduced activity levels, which can lead to weight gain. The long undiagnosed period of sarcoidosis also complicates retrospective studies, as it limits the ability to assess pre-existing factors. Nevertheless, evidence from prospective studies such as the Black Women’s Health Study has shown that the risk of sarcoidosis increases with higher BMI and weight gain [41]. Similar findings have been reported in other longitudinal studies involving predominantly white women of European descent [42].

Tobacco use also presents an intriguing connection to sarcoidosis. Studies including the ACCESS study suggest that smokers may have a lower risk of developing the disease [33,43,44]. Nicotine, known to modulate immune responses in animal models, can suppress granulomatous inflammation via restoring and augmenting Treg cell activity. A small randomized, double-blind, placebo-controlled study with 50 sarcoidosis patients explored the effects of nicotine patches over 24 weeks. The treatment was well tolerated and led to a slight but statistically significant improvement in lung function (FVC) [45].

A link of sarcoidosis with rurally linked exposures and living on a farm has been described. Apart from inhaled exposure, this also pertains to dietary correlates such as drinking from non-public water supplies (wells) [46]. Such a kind of diet could enrich gut microbial flora via the known gut–lung axis impact on lung microbiome in sarcoidosis. Atopobium, fusobacterium or other bacilli have been found in bronchoalveolar lavage fluid of sarcoidosis patients [47]. Granulomatous diseases of the gastrointestinal tract that mimic sarcoidosis, Crohn’s disease or Whipple disease, suggest pathogenetic and dietary similarities [48].

### 2.5. Seasonal Exposure

Some studies have tried to evaluate the correlation between sarcoidosis and seasonal exposure. In a retrospective cohort study which took place in Olmsted Country in Minnesota, it was established that incident sarcoidosis was less common in autumn than in winter and summer [49]. However, in another retrospective study in the U.S., no seasonal influence was found. In a single-center retrospective study from India, a higher incidence in the summer months was noticed [50]. An increased summer incidence is supposedly attributed to increased travelling and environmental exposure in younger or middle-aged individuals. Seasonal variation may be due to geographical differences.

Possible risk factors of sarcoidosis are presented in Table 1.

## 3. Immunopathogenesis

Sarcoidosis’ histopathological profile is characterized by the occurrence of non-caseating epithelioid cell granulomas, being described as a product of immunological hypersensitivity [52]. Typically, the granulomas of sarcoidosis feature a central core of multinucleated giant cells, derived from macrophages differentiated to epithelial cells, and CD68^+^ macrophages of the so-called M1-like pro-inflammatory phenotype. These are surrounded by an outer ring formed mostly by CD8^+^ T cells, Tregs, B cells and fibroblasts [52,53]. Most importantly, the outer ring consists of CD4^+^ T helper cells, especially Th1 and Th17.1 [54].

Granuloma formation begins when the phagocytosis of a substance is executed by an antigen-presenting cell (APC), most commonly a macrophage or a dendritic cell (DC), and it presents catabolized parts of it on its membrane via the class II MHC (Major Histocompatibility Complex) [55]. *HLA-DRB1* variants are associated with different sarcoidosis phenotypes, as discussed in the Genetic Background section. 

Propionibacterium acnes is a factor commonly implicated in the formation of sarcoidosis granulomas. In mice, antimicrobial treatment of Propionibacterium acnes decreased sarcoidosis granulomas [56]. Case reports have suggested an effect of antimicrobial treatment in humans but no randomized controlled trial has confirmed these results [57].

The function of APCs in sarcoidosis is mostly triggered through Toll-like receptors (TLRs) and the pathways that they regulate. TLRs belong to the broader family of pattern recognition receptors (PRRs), which are critical for the recognition of microorganisms [58]. Genetic variations in TLRs have been associated with increased risk of sarcoidosis [1,58]: a genetic deficiency affecting the MyD88 protein results in increased burden of Propionibacterium acnes and larger granulomas in animal models of sarcoidosis, while its polymorphisms have been associated with increased risk of sarcoidosis in humans [21,59]. Larger granulomas were also observed in mice lacking the *CybB/Nox2* gene when exposed to Propionibacterium acnes. CybB/Nox2 is implicated in bacterial clearance through reactive oxygen species [59]. Other TLR polymorphisms have been associated with chronic sarcoidosis [19,20,60].

Maladaptive trained immunity of alveolar macrophages (AMs) is implicated in sarcoidosis [61,62]. Certain stimuli, such as bacillus Calmette–Guerin (BCG), have been proven to activate designated functional programs of trained immunity pathways and can lead to sensitization against multiple antigens and diseases [62,63]. Monocytes’ progenitors mainly and subsequently AMs receive abnormal epigenetic and metabolomic programming due to trained immunity in genetically predisposed patients [61]. As a result, they respond irregularly to certain triggers, which could lead to granuloma formation [61]. T cells recognize the antigen via its TCR receptor, forming a trimolecular complex with the MHC and the antigen. Followingly, oligoclonal CD4+ T-cell differentiation and expansion occur dependently on the cytokines present in the inflammatory microenvironment, most of which are being secreted by the APCs themselves. Such cytokines include IL-1β, IL-12, IL-6 and transforming growth factor beta (TGF-β) [55,64]. In that manner, the APCs have control over the course of the immunological response. TGF-β has an anti-inflammatory effect. Increased levels of TGF-β released by AMs are associated with spontaneous remission in sarcoidosis [65]. TGF-β polymorphisms, however, have been implicated in sarcoidosis-associated fibrosis [66]. From this point on, the T cells are the ones pathologically dictating the creation of the granuloma, releasing the highly granuloma-associated conception cytokines IFN-γ, IL-17A and IL-2, leading to the circulation of other cell types [64]. 

Dysregulated autophagy is essential in sarcoidosis persistence. Both animal models and human trials indicate the overactivation of the mTOR and JAK/STAT pathways in M1 macrophages, regulating autophagy procedures [67,68]. The overactivation of the mTOR pathway favors the formation of multinucleated giant cells and has been associated with chronic sarcoidosis progression [67,69]. Familiar sarcoidosis has been associated with mutations in the mTOR regulators [70]. mTOR pathway activation has also been connected to the secretion of IL-1β and IL-6 in the granuloma microenvironment [67]. A 17-gene signature involving the JAK/STAT pathway has been associated with increased sarcoidosis severity [71].

The most feared complication of sarcoidosis, fibrosis, appears to be utterly connected with the macrophages’ behavior in the granuloma, with an unexpected shift in their phenotype from M1 to M2. This shift has not been fully explained yet [72]. The way the model of M1/M2 phenotypes is presented, however, is quite simplified, with in vivo data suggesting that macrophages function in a broader spectrum, of which the M1 and M2 phenotypes are the polar opposites [73]. Mostly, the macrophages’ function is dictated by the cell’s microenvironment triggering its TLRs [73].

Similarly to macrophages, DCs are involved in important pro-inflammatory cytokine secretion, mainly IL-12, contributing to the polarization of CD4^+^ T cells in the Th1/Th17 spectrum towards the Th1 pro-inflammatory phenotype [74]. Increased levels of IL-12p40 have been associated with active pulmonary sarcoidosis [75]. The differentiation is dictated by the consistency of the local milieu in cytokines, with IL-12 and IFN-γ contributing to the occurrence of the Th1 phenotype and of IL-6, IL-23 and TGF-β of the less aggressive Th17 phenotype [76]. The maturity of DCs apparently has a role in sarcoidosis’ immunological profile: higher rates of DCs with high expression of HLA-DR have been measured in the BALF of patients with sarcoidosis [77].

### T Cells in Sarcoidosis Immunopathogenesis

T helper CD4^+^ cells have the most central role in granuloma formation [64]. Sarcoidosis is characterized by an “immune paradox” [64]: a hypersensitivity reaction is present in areas of chronic inflammation, such as the mediastinal lymph nodes (MLNs) or the lungs, while sarcoidosis patients present peripheral anergy, and they are prone to other infections, mostly attributed to the lower concentration of CD4^+^ in peripheral blood compared with healthy controls. The opposite is found in the BALF of sarcoidosis patients [78]. Moreover, the CD4^+^ cells available in the peripheral circulation also show immunological exhaustion [79].

The CD4^+^ typing spectrum has four cell categories: Th1; Th17.1; double producing Th cells (DP), which are considered to be a proliferative type of Th17.1 [80]; and Th17 cells [81]. Two phenotypes of the T helper cells are crucial in sarcoidosis: Th1 and Th17.1 cells. In chronic sarcoidosis, a high inflammatory response of Th1 and Th17.1 cells is seen, while increased numbers of Th17 cells have been associated with Löfgren syndrome [1,80,82]. 

The polarization of this T-cell type is dictated by the milieu’s consistency in cytokines but also by certain receptors and transcription regulators being expressed by the cell, most importantly the receptors CXCR3, CCR4 and CCR6 and the factors T-bet and RORγT [18]. CXCR3 is affiliated with Th1 and CCR6 with the Th17 lineage [78]. The factors T-bet and RORγΤ are expressed by cells of the Th1 and Th17 lineages, respectively, while Th17.1 cells express both factors [83]. Th1 cells produce IL-2, TNF-α and most importantly IFN-γ. IFN-γ is also produced by Th17.1 cells [81]. Increased expression of IL-2 and TNF-α has been associated with chronic active sarcoidosis and reduced expression with Löffler syndrome [84]. IFN-γ is a main cytokine in the pathogenesis of sarcoidosis, appearing in higher rates in patients´ BALF, especially in those who develop chronic disease [64,84]. Th17 cells mostly produce IL-17A, while Th17.1 cells were recognized as the main producers of IFN-γ in the BALF of patients with chronic sarcoidosis [64]. The simultaneous production of the regulators T-bet and RORγT has been found in Löfgren syndrome, acute sarcoidosis with immediate resolution after glucocorticoids, combined with low levels of IFN-γ, which is probably attributed to DP cells [83]. On the contrary, patients with higher levels of Th1 and Th17.1 cells, which produce the multidrug resistance protein 1 (MRP-1), did not improve after treatment with glucocorticoids, thus supporting the association of Th1 and Th17.1 cells with chronic disease [85].

Another factor possibly promoting sarcoidosis’ immunopathogenesis is the decreased capacity of Treg cells. An increased proportion of Tregs was detected in peripheral blood samples of sarcoidosis patients [78]. On the other hand, Sakthivel et al. reported Treg suppression being modulated in Löfgren syndrome patients [86]. The larger portion of the literature, however, reports a decreased immunosuppressive capacity pertaining to Treg function [87,88]. The study by Oswald-Richter et al. highlighted the Treg role in sarcoidosis relating Treg cell activity restoration with the disease’s resolution [87]. 

The complex interplay between different T-cell populations has been recently reported to correlate to different phenotypes. By using flow cytometry, the different T-cell populations of the peripheral blood of patients with sarcoidosis were analyzed: decreased CD4^+^ T cells and increased Tregs and CD8^+^ γδ T cells correlated with worse prognosis, while naïve CD4^+^ T cells displayed an activated phenotype with increased CD25 expression in patients with active chronic disease. Chronic sarcoidosis was associated with a distinctive Treg phenotype showing high expression of CD25, CTLA4, CD69, PD-1 and CD95 [89].

The role of other inflammatory cells in sarcoidosis immunopathogenesis is under intensive research, including natural killer (NK) cells [90,91] and B cells [92,93], with studies suggesting possible interplays between their function and CD4^+^ T cells [94]. NK-cell subsets CD56 and CD16 have been detected in BALF, secreting increased levels of TNF-α and IFN-γ, compared with healthy controls, subsequently probably contributing to Th1 alveolitis [90]. Also, amplified NK-cell percentages in BALF were corelated with a worse disease prognosis [95]. Pertaining to B cells, a decreased memory/naïve T-cell ratio has been observed in sarcoidosis, accompanied by B cells surrounding the granuloma [96]. Additionally, studies have correlated IL-10-producing B-cell presence and increased serum B-cell activating factor (BAFF) with chronic sarcoidosis [62,93].

A schematic presentation of basic events in the pathogenesis of sarcoidosis is shown in Figure 1.

## 4. Prognostic Factors in Sarcoidosis

The prognostic factors in sarcoidosis are particularly important for determining the clinical outcome of the disease and the best possible treatment and to fully elucidate the risk factors to prevent or even avert the disease in its early stages. Prognosis depends on multiple determinants, such as clinical and imaging factors (e.g., Scadding radiography stages), biomarkers, demographic factors (e.g., socioeconomics, sex and race), genetic factors (e.g., human leukocyte antigen class II and tumor necrosis factor-a), and others.

### 4.1. Demographic Factors

Sarcoidosis can develop regardless of an individual’s demographic profile, although its prevalence and clinical manifestations vary based on factors such as sex, age, socioeconomic status, race and geographic distribution. African Americans and Scandinavians are more likely to develop sarcoidosis than the rest of the Caucasian population. The most common age for the appearance of sarcoidosis is under 50 years old, with 70% of the cases being from 25 to 40 years and a second peak for women over 50 years old [97].

A 5-year study conducted in the United States (U.S.) by a Health Maintenance Organization describes the annual incidence of sarcoidosis among African Americans to be 35.5 per 100.000 with lifetime risk of sarcoidosis at 2.4%, while the same figures for Caucasian Americans are 10.9 and 0.85%, respectively (age-related adjustments were made) [98]. In a study conducted in the U.S. that did not take socioeconomic factors into consideration, a higher sarcoidosis-related mortality rate was observed in Black individuals, with 3 deaths per million for males and 10 deaths per million for females, compared with 1 death per million in White individuals. In the same study, the authors noted a decrease in life expectancy between individuals with and without a sarcoidosis background, with the average ages at death being 57.2° ± 1.8 and 71.4 ± 1.1, respectively [99].

In A Case–Control Etiologic Study of Sarcoidosis, it was highlighted that having low income, being female, being Black, having a low education level (defined as not beyond high school), having state insurance or no insurance, having difficulties in obtaining medication, having barriers to care and having missed one or more medical appointments over the last six months are associated with sarcoidosis severity at presentation. In the same study, low socioeconomic status alone (after adjustments for race, sex and education) was corelated with worsening sarcoidosis [100]. Geographic heterogeneity also affects the prognosis in sarcoidosis. In Western countries, the most common cause of sarcoidosis deaths is pulmonary fibrosis that causes pulmonary hypertension, respiratory failure or both [93], and only a small percentage of deaths are due to central nervous system and cardiac sarcoidosis (CS) or portal hypertension [97]. These results are in opposition to a study conducted in Japan, where the main cause for sarcoidosis deaths (77%) was cardiac involvement [101].

### 4.2. Clinical Factors

Longitudinal studies with sample sizes that vary between 215 and 1774 suggest that some clinical characteristics of the patients, such us erythema nodosum, stage 1 disease, Löfgren syndrome and resolution or improvement in disease, are linked to a more favorable prognosis over a follow-up period ranging from 2 years to 14 years [95,96,97,98]. Other studies showed that more extensive pulmonary involvement and extra-pulmonary organ involvement are correlated with a poorer prognosis [99,102,103,104,105,106].

A large Swedish observational cohort study, which included 8207 sarcoidosis patients and 81,119 age- and sex-matched individuals from the general population, found that a diagnosis of sarcoidosis was associated with a mortality rate of 11 per 100 person-years over a median follow-up of 5.9 years. In comparison, the mortality rate among controls without sarcoidosis was 6.7 per 100 person-years during the same period. The elevated mortality in sarcoidosis patients requiring treatment within the first 3 months of diagnosis is the primary reason for the disparity in mortality between sarcoidosis patients and controls. These findings suggest that symptomatic sarcoidosis patients are at the highest risk of mortality [107].

Regarding other sarcoidosis phenotypes, some common specific cutaneous sarcoidosis findings provide insights into the progression of the disease. Papules and papulonodular lesions are the most common morphology types of the specific cutaneous manifestations of sarcoidosis associated with a favorable prognosis. Plaques are associated with a chronic disease, and lastly, lupus pernio is associated with a chronic and refractory type of the disease [108,109]. Concerning cardiac sarcoidosis, Cardiac Magnetic Resonance Imaging (CMRI) is the cornerstone of diagnosis of CS with both high sensitivity and specificity (>90%), allowing for the identification of areas of damaged myocardium with the utilization of Late Gadolinium Enhancement (LGE). LGE on CMRI is also a very useful prognostic tool allowing for risk stratification in cardiac sarcoidosis and poor prognosis assessment, since LGE is associated with death and ventricular tachycardia but also with the size of the granulomatous infiltrate [110,111,112,113,114,115].

### 4.3. Biomarkers

Several biomarkers have been studied in sarcoidosis. A study contacted by Elena Bargagli et al. with a sample of 232 sarcoidosis patients analyzed chitotriosidase as a potential prognostic biomarker [116]. They found that those with the highest serum chitotriosidase concentration were symptomatic patients with a refractory type of disease on steroids and functional decline in the last year. In these patients, the increase in the steroid dosage and the use of a new immunosuppressive agent led to a reduction in the enzyme. Moreover, another study suggested that serum chitotriosidase is linked with extra-pulmonary localizations [117]. A cohort study, with a total of 30 participants, by Paone G. et al. found a combination of six markers from BAL that predicted pulmonary function worsening in 96% of patients. The formula was then also validated on a group of nine sarcoidosis patients with the result of 77.8% correct classification (95% with a confidence interval of 45.3–93.7%) [118]. Additionally, an oxidative DNA damage marker, urinary (U) 8-hydroxy-2’-deoxyguanosine (8-OHdG), was investigated in two studies conducted in Japan [119,120]. In these studies, individuals with cardiac sarcoidosis (n = 30) were classified as active or non-active based on 18F-fluorodeoxyglucose positron emission tomography/computed tomography (18F-FDG PET/CT) imaging. The results demonstrate that 8-OHdG is an important biomarker for assessing inflammatory activity and therapeutic response to corticosteroid and for predicting cardiovascular-related death in patients with CS. These findings suggest that high levels of 8-OHdG in active cardiac sarcoidosis patients are correlated with resistance to corticosteroid therapy. An association between cardiac troponin I and fatal arrythmia (HR 2.418, *p* = 0.003) in CS patients was described by Takatoyo Kiko et al. [121]. In their study, they compared multiple biomarkers between noncardiac sarcoidosis patients (n = 123) and cardiac sarcoidosis patients (n = 49). In another study, Bergantini L. et al. found that Krebs von den Lungen-6 (KL-6) with a cut-off value of 303.5 IU/m (AUC = 0.79, 78%, 85%) can be used effectively as a biomarker for fibrotic lungs which is correlated with poor survival [117]. Serum-soluble interleukin 2 receptor (sIL-2R) is a prognostic biomarker that has been linked to the development of chronic sarcoidosis, as demonstrated in a retrospective cohort study that included 121 patients with sarcoidosis and 70 healthy controls [122]. Serum angiotensin-converting enzyme (ACE) has long been associated with sarcoidosis diagnosis, monitoring and activity. However, its accuracy is decreased by intrinsic polymorphisms, with a sensitivity and specificity in the area of 50–70%. In a study of 823 patients in China, serum ACE and IL-2R were useful in assessing initial extra-pulmonary involvement and monitoring activity after treatment initiation [123,124]. One last biomarker examined in sarcoidosis is BALF cellular and sub-populations analysis. An association was found between increased BALF percentage of NK cells in sarcoidosis and a poor prognosis with a higher likelihood of treatment with steroids, extra-pulmonary disease and Scadding stage 4 [95,125]. Lymphocytic BALF correlates with increased disease activity, while neutrophilic BALF is associated with severe chronic disease [126]. Lastly, a worse prognosis is correlated with increased activated naïve T cells in the circulation [89].

### 4.4. COVID-19

Apart from the traditional prognostic factors, COVID-19 impact on the course of sarcoidosis has been reported. This is due to two factors: firstly, because of initially reported negative outcomes of sarcoidosis patients and COVID-19 infection and disease, and secondly, because of shared pathogenetic mechanisms. These mechanisms, briefly, include angiotensin-converting enzyme (ACE) receptors in alveolar epithelial cells, disrupted autophagy including the JAK-STAT and mTOR pathways and associated gene sequencing [68,127,128]. It is now known, however, that COVID-19 infection is not increased in sarcoidosis. COVID-19 severity and prognosis in sarcoidosis depend on underlying disease severity, with individuals more prone to negative outcomes being advanced in age, presenting comorbidities and severe pre-existing pulmonary or extra-pulmonary disease and being treated with certain immunosuppressants (rituximab and high-dose prednisolone) [129,130].

Rarely, granulomatous lesions have been observed in various organs after COVID-19 infection (skin and lymph nodes) or, interestingly, after COVID-19 vaccination [131]. Cutaneous granulomas or Löfgren syndrome have been reported after second-dose vaccination with Moderna mRNA-1273 or ChadOx-1 and Astra Zeneca and erythema nodosum after the second dose of the Pfizer-BioNTech mRNA vaccine [131].

## 5. Phenotypes in Sarcoidosis

Sarcoidosis, by definition a multi-organ disease, may present with numerous manifestations, mimic a variety of diseases and exhibit various prognosis and clinical outcome status ranging from self-resolution to death [132]. The disease is so heterogeneous that organ-specific disease and patient-reported outcomes considerably vary [133]. Autopsy studies could drastically alter perceived phenotypes in sarcoidosis by detecting asymptomatic organ involvement. For the above reasons, attempts have been made to classify sarcoidosis in phenotypes, i.e., groups with several similar characteristics. The purpose of phenotyping sarcoidosis may be educational, with stratification by severity risk, or phenotyping could have prognostic, research and therapeutic purposes. 

### 5.1. Radiology Phenotyping

Historically, Karl Wurm and J.G. Scadding proposed the first phenotyping of sarcoidosis based on plain chest radiography. The Scadding radiographic stages (0 to 4) are partly associated with the prognosis and mortality [134]. The radiographic findings for each Scadding stage 0–4 are as follows: normal (0), bilateral hilar lymphadenopathy without pulmonary infiltrates (1), bilateral hilar lymphadenopathy with pulmonary infiltrates (2), pulmonary infiltrates without bilateral hilar lymphadenopathy (3) and extensive fibrosis with distortion or bullae (4). Five longitudinal studies with pulmonary sarcoidosis patients from the United States (244 patients), the United kingdom (818 patients), Denmark (210 patients), France (142 patients) and Sweden (505 patients) followed up for periods between 1 and 15 years found 49–82% radiographic resolution for stage 1 disease, 31–68% for stage 2 disease, 10–38% for stage 3 disease and 0% for stage 4 disease. Furthermore, the mortality rates for stages 0–4 were 0–9%, 5–11%, 12–18% and 16–17%, respectively. In addition, the percentages of patients that will have a pulmonary obstruction (assessed by the ratio of FEV1 to FVC) differ: 0% of the patients in stage 0, 6% of the patients in stage 1, 13% of the patients in stage 2 and 16% of the patients in stages 3 and 4 [6,101,106,135,136,137,138,139,140]. The Scadding classification offers a risk assessment but does not correlate efficiently with patient symptoms, pulmonary function tests, computed tomography (CT) findings, specific pulmonary complications, need for treatment and extra-pulmonary disease.

Thoracic High-Resolution CT (HRCT) is able to more accurately stage pulmonary sarcoidosis. In their study of 227 Chinese and American patients, Zhang et al. reported discordancy between chest radiography and HRCT in 50% of cases. HRCT was able to detect pleural involvement and had a significant negative correlation with carbon monoxide diffusing capacity (DLCO) [141]. HRCT is also found to correlate with sIL-2R as per disease activity [142]. Furthermore, HRCT is able to detect vascular and parenchymal complications and comorbidities, like coronary heart disease, but, importantly, recognizes mortality risk in sarcoidosis. In a study of 452 cases, Kirkil et al. found that apart from older age and pulmonary hypertension, an extent of fibrosis >20% on HRCT is an independent predictor of mortality [135]. A recent multinational Delphi consensus study reported two phenotypes, non-fibrotic and likely to be fibrotic [143]. HRCT may safely recognize pulmonary fibrosis and complicated pulmonary sarcoidosis, as illustrated in Table 2 [144].

### 5.2. Pulmonary Function Phenotyping

Pulmonary function test (PFT) studies have shown mixed results in sarcoidosis. PFTs may be normal or exhibit an obstructive, a restrictive or a mixed pattern or an isolated DLCO reduction [145]. Two recent studies have shed light on PFT phenotyping and significance in sarcoidosis. 

Kouranos et al. studied 1110 patients with pulmonary sarcoidosis [145]. The prevalence of a mixed ventilatory defect defined via total lung capacity was 10.4% in the whole cohort, rising to 25.9% in patients with airflow obstruction. Compared with isolated airflow obstruction, mixed defects were associated with lower DLCO (50.7 ± 16.3 vs. 70.8 ± 18.1; *p* < 0.0001), higher prevalence of chest radiographic stage 4 disease (63.5% vs. 38.3%; *p* < 0.0001) and higher mortality (hazard ratio, 2.36; 95% CI, 1.34–4.15; *p* = 0.003). Mixed disease, therefore, is present in approximately 25% of patients with pulmonary sarcoidosis and airflow obstruction and is associated with lower DLCO, stage 4 disease and higher mortality than seen in a purely obstructive defect [145].

Sharp et al. performed spirometry and DLCO testing in 562 patients with pulmonary sarcoidosis and found significant differences by sex and race [146]. They reported that among Black individuals, the most common pattern was the restrictive one, while White individuals had most commonly normal PFTs (66%). Males frequently showed obstruction and females restriction. The authors concluded that forced expiratory volume in 1 s (FEV1) and forced vital capacity (FVC) are affected by race and disease duration and while DLCO by race, male sex, disease duration and smoking [146].

### 5.3. Clinical Tools for Phenotyping

A sarcoidosis assessment instrument was developed by the Steering Committee of A Case Control Etiologic Study of Sarcoidosis (ACCESS). The purpose of this tool was to describe organ involvement in biopsy-proven sarcoidosis and guide clinicians in accurately phenotyping sarcoidosis based on symptoms, signs, biopsy data and ACCESS study findings, as well as offering guidance as to whether a new biopsy should be performed or not based on the likelihood of sarcoidosis [147]. An updated organ assessment tool was published by the World Association of Sarcoidosis and Other Granulomatous diseases (WASOG) in 2014 [148]. This tool aimed to incorporate in clinical practice the latest developments in the diagnosis of cardiac sarcoidosis (FDG-PET and cardiac MRI) and may not provide prognostic information. 

In recent years, clinical tools which incorporate patient-reported outcomes (PROs) have been developed and have been used widely in research to assess the response and efficacy of novel treatments. Such tools cluster patients more efficiently than the traditional clinical tools mentioned above, with implications for research purposes where organ clustering and omics may allow for the better endotyping of this heterogeneous disease. Such tools are the Sarcoidosis Assessment Tool (SAT), which incorporates physiology with PROs and patient global assessment visual scales, and King’s Sarcoidosis Questionnaire, which incorporates solely PROs based on pulmonary, eye and skin symptoms [149,150].

### 5.4. Nuclear Imaging Phenotyping

The role of fluorine-18-fluorodeoxyglucose positron emission tomography (FDG-PET) is extensively studied in sarcoidosis. Currently, it is only approved for the diagnosis of cardiac sarcoidosis. It has shown the benefit of the recognition of indolent disease, new organ involvement, response to treatment and disease activity in the event of pulmonary fibrosis [151,152].

FDG-PET is increasingly used to stratify patients and monitor disease activity [152,153]. However, its role in phenotyping, organ clustering and prognosis needs further clarification. Papiris et al. used FDG-PET in clinical practice in 195 sarcoidosis patients, in order to identify novel phenotypes and cluster patients based on FDG-PET findings. They described four phenotypes: thoracic nodal hilar–mediastinal, thoracic nodal hilar–mediastinal and lungs, an extended thoracic and extra-thoracic only nodal phenotype including inguinal–abdominal–supraclavicular stations, and all the above plus systemic organs and tissues such as muscles–bones–spleen and skin [154]. Organ clustering without genetic background and further endotyping, however, may not explain the heterogeneity and the diverse disease progression outcomes in sarcoidosis [153]. 

### 5.5. Genetic Phenotyping

The GenPhenReSa consortium is an ambitious multinational collaboration that illuminates a new paradigm for sarcoidosis research aiming to elaborate on genotype influence on phenotype diversity in sarcoidosis across Europe [155]. A total of 2163 Caucasian patients with sarcoidosis were phenotyped at 31 study centers according to a standardized protocol. Patients with acute onset were mainly female, young and of Scadding type 1 or 2. Female patients showed a significantly higher frequency of eye and skin involvement and complained more about fatigue. Based on multidimensional correspondence analysis, the authors performed unsupervised hierarchical clustering on principal components to identify novel clinical associations. Five phenotypic organ-based clusters emerged from the analysis, and patients could be clearly stratified into five distinct yet undescribed subgroups according to predominant organ involvement: (1) abdominal organ involvement, (2) ocular–cardiac–cutaneous–central nervous system disease involvement (OCCC), (3) musculoskeletal–cutaneous involvement, (4) pulmonary and intrathoracic lymph node involvement and (5) extra-pulmonary involvement. Classification in a given cluster entails higher odds for certain other clinical features, such as acute versus subacute onset, symptoms and need for therapy. Similar organ clustering is also reported in other studies. In the Sarcoidosis Genetic Analysis (SAGA) study in African-Americans, liver, spleen and bone marrow involvement segregated into a phenotypic cluster [156]. 

The authors, in a subsequent publication, reported two polymorphisms, one in *TNF- 308G/A (rs1800629*) and an HLA polymorphism (*rs4143332*), associated with acute-onset disease [157]. Multi-locus models with sets of three SNPs in different genes showed strong associations with the acute-onset phenotype in Serbia and Poland, demonstrating potential region-specific genetic links with clinical features and associations with environmental risk factors. Apart from acute-onset disease, no genotype–phenotype association uniformly present in the whole European cohort was observed, when analyzed with an appropriate meta-analysis, although different unique associations were present in different geographic sub-cohorts. These findings taken together indicate the importance of epigenetics and transcription factors in individuals susceptible to the development of a specific phenotype.

Different phenotyping methods are presented in Table 3.

## 6. Endotypes in Sarcoidosis

The holy grail in sarcoidosis research is the identification of determining factors for disease progression to fibrosis, in other words, the clinical, gene and transcription profiles (endotypes) associated with resolution or dismal prognosis to fibrosis. As discussed above, genotype–phenotype associations are necessary to navigate through disease diversity to distinct outcomes that may be the target of future therapies.

### 6.1. Genotype Analysis

Genetic factors are important in assessing the progression of sarcoidosis. As shown in a multi-omics study conducted by Konigsberg I.R. et al., there are some genetic features, such as Differentially Expressed Genes (DEGs), methylation, mRNA and miRNA, that can be used as prognostic tools in sarcoidosis [158]. Firstly, in their single-omics approach, they identified four DEGs (*IGHV3-72*, *MIR4640*, *SEPP1* and *CPB2*) by comparing progressive and nonprogressive patients, but all of them, except SEPP1, were expressed at very low levels, making any observed changes harder to detect or less meaningful, displaying small effect sizes (|log2(fold change)| < 0.01). Similarly, a multi-omics integrated model demonstrated that miR-146a3p and miR378 were upregulated in progressive sarcoidosis, and novel genes, such as *IL-20RB*, *ABCC11*, *SFSWAP* and *AGBL4*, showed affinity for progressive pulmonary sarcoidosis [158]. Three miRNAs (miR-21-5p, miR-340-5p and miR-212-3p) differed between patients with and without Löfgren syndrome, and they were associated with TGF-*β* signaling, while in another study, a higher number of miR-155, let-7c transcripts and transcription factor T-bet was associated with progressive disease [159,160]. 

The GRADS study analyzed the BALF genome-wide transcriptome with clinical traits in 215 sarcoidosis patients [161]. Among the CT phenotypic traits, reticular abnormality, hilar lymphadenopathy and bronchial wall thickening were significantly associated with a large number of genes. Genes increased in the presence of hilar lymphadenopathy were enriched for T helper type 1 (Th1) and Th17, interferon (IFN)-γ and nuclear factor of activated T-cell (NFAT) signaling. The genes reported in the analysis were *CD28*, *STAT1*, *CXCR3* and *CCR4*. The genes increased with more severe bronchial wall thickening were enriched for the aberrant IL-2 and IL-7 pathways, including *MRC2*, *SLC40A1*, *F2R*, *IL7*, *PTPN7*, *ADORA2A*, *SPRY2*, *PLA2G7* and *PTGS1*. Reticular abnormalities on CT scan were positively correlated with fibrosis-associated genes such as *TGFBR1*, *COL3A1*, *TLR3*, *ID1*, *TCF4*, *IGFBP6*, *PLA2G7*, *FADS1*, *ARGHAP12* and *MMP10*. In conclusion, GRADS investigators described eight novel phenotypes, namely, multi-organ, stage 1 untreated, stage 2–3 treated and untreated, stage 4 treated and untreated, acute untreated and remitting untreated… Finally, they were able to cluster PFTs, age, gender, Scadding stage and BALF cells according to four novel endotypes: chronic sarcoidosis, hilar lymphadenopathy and acute lymphocytic inflammation, multi-organ involvement with increased immune response and extra-ocular involvement with PI3K activation.

Genes associated with the Major Histocompatibility Complex (MHC), also referred to as human leukocyte antigen (HLA), appear to be useful in multiple aspects of sarcoidosis prognosis, such as disease risk, target-organ phenotypes and clinical outcomes. Among the HLA II genes, a prominent representative is the *HLA-DBR1* locus, whose alleles have been linked with different sarcoidosis phenotypes and disease progression [162]. It should be noted here that transcriptomics differentiates among sarcoidosis lymph nodes and pulmonary parenchyma. As shown in the study by Casanova et al., downregulated genes vary between lymph nodes and sarcoidosis lungs. Differences were delineated by a clear immunological response, involving leucocyte migration and neutrophil chemotaxis in the lymph nodes, while a structural regenerative response, characterized by cell migration and angiogenesis, was observed at the lung level, leading to the conclusion that transcript genes are disease- and tissue-specific [163].

A meta-analysis conducted by Lin Y et al. highlighted the connection between the polymorphism *rs2076530* of the Butyrophilin-like protein 2 (*BTLN2*) gene and sarcoidosis susceptibility [164]. Additionally, polymorphisms in the *ANXA11* gene are associated with varying risk of developing sarcoidosis, as well as with the chronicity of the disease. Specifically, regarding the *ANXA11* locus *rs1049550*, studies have shown a protective effect by the minor T allele. Moreover, other variants have been linked with chronic disease, pulmonary fibrosis, advanced Scadding stages and non-Löfgren syndrome [165,166,167,168,169,170,171,172]. These findings, combined with the fact that *ANXA11* is among the few genes specifically associated with sarcoidosis, suggest that more research must be conducted to fully understand the disease and distinguish it from other immune-mediated diseases [172]. Serum TNF-α levels are linked to an endotype of the disease characterized by radiologically progressive sarcoidosis with impaired pulmonary function and exercise ability impairment. A cut-off value of 28.58 pg/mL was set to differentiate patients with active, progressive sarcoidosis from those with nonprogressive disease [173].

### 6.2. Autophagy

As already reported previously, sarcoidosis is characterized by oligoclonal T-cell expansion and a Th-1 inflammatory response leading to granuloma formation. This is mediated by cytokines such as TNF-α, IFN-γ and IL-1β. Further contributing factors to granuloma persistence are T-cell exhaustion and regulators of innate immunity. These include Th-17 activation and T-regulatory-cell exhaustion. 

Dysregulated autophagy through persistent m-TORC-1 pathways, persistent JAK-STAT signal transduction pathways and Granulocyte Macrophage Colony-Stimulating Factor (GM-CSF) is essential in sarcoidosis persistence and faulted antigen clearance. Autophagy is a phago-lysosomal activity to maintain cellular homeostasis by the clearance of intracellular pathogens and molecules. Initially, it was found in in vivo studies that the activation of the m-TOR pathway in Tuberous Sclerosis-2 gene-deficient mice led to the development of sarcoid granulomas [69]. This has been subsequently described not only in human in vitro models but also in genetic studies, which have shown specific gene polymorphisms not only in the m-TOR and JAK genetic locations but also in Transforming Growth Factor-B3 (TGF-β3) and Th-17, which are most associated with various sarcoidosis endotypes [174]. Whole-exome sequencing in French familial sarcoidosis cases demonstrated autophagy- and mitophagy-related pathway genes [68].

Endotypes in sarcoidosis are presented in Table 4.

### 6.3. Implications for Treatment

It is clear that the characterization of phenotypes and endotypes will lead to orthological treatment selection, robust organization of clinical trials, precision medicine and better outcomes. An example of this is the OCCC phenotype in the GenPhenReSa study. The OCCC phenotype involves organs that are known as responders to tumor necrosis factor antagonists [175]. Recent results have shown the effectiveness of mTOR inhibitors and JAK inhibitors in sarcoidosis. In detail, sirolimus has shown sustained efficacy in 10 patients with cutaneous sarcoidosis [53]. On the other hand, the JAK inhibitor tofacitinib was effective in chronic pulmonary and cutaneous sarcoidosis [176]. The results of the aforementioned studies remain to be confirmed in larger placebo-controlled studies. It is anticipated that advanced endotyping and gene sequencing will permit more targeted treatment trials in the future. Aspects connecting endotypes and phenotypes in sarcoidosis are presented in Figure 2.

## 7. Conclusions

Despite the progress made in our understanding of sarcoidosis, several aspects still remain unclear. Sarcoidosis is still considered a disease of unknown etiology, occurring in patients exposed to an unknown antigen with a genetic predisposition. The result is uncontrolled granuloma formation. Macrophages and Th1/Th17 cells play a very important role not only in granuloma formation but also in the perpetuation of inflammation and progression to fibrosis. New aspects are autophagy disturbances and Treg exhaustion. Since sarcoidosis can affect any organ, great efforts have been made to better categorize patients with the disease, in order to achieve personalized treatment with few side-effects. Phenotyping methods evolve with the development of clinical tools. HRCT and PET-CT provide new insights into different phenotypes, with variable disease course. Genotyping studies, such as the GRADS study, provide valuable information about the genetic basis of sarcoidosis manifestations. Endotyping allows for patient-tailored treatment, as the OCCC phenotype in the GenPhenReSa study and the favorite response to TNF-α antagonists or JAK-inhibitors in chronic pulmonary sarcoidosis. It is anticipated that single-cell profiling and the efficient endotyping of the disease with more outcomes (cardiac disease, pulmonary vascular disease and fibrosis) will allow us in the future to predict the more deleterious course of the disease, the disease that will lead to persistent fibrosis and end-organ damage. The prediction of disease behavior according to endotyping and phenotyping may allow us to tailor therapies in a more efficient manner. Tailored therapies according to disease prediction, earlier on in the disease course and with minimal use of potentially harmful treatments may lead to better outcomes compared with the ones achieved with the first-line agents that are currently used.

## Figures and Tables

**Figure 1 biomedicines-13-00287-f001:**
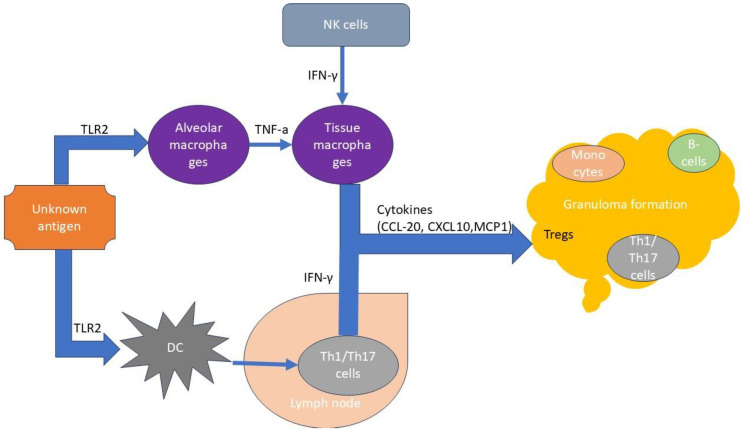
Basic events in the pathogenesis of sarcoidosis. An unknown antigen activates, through TLRs, alveolar macrophages and dendritic cells. Macrophages aggregate and transform into multinucleated giant cells. Natural killer cells significantly contribute to this process. The antigen-presenting cells (dendritic cells and macrophages) lead to the oligoclonal expansion of the Th1/Th17 lymphocytes, which play a cardinal role in granuloma formation. Treg cell exhaustion presents in lymph nodes. CCL-20: C-C motif ligand 20; CXCL10: C-X-C cytokine motif ligand 10; DC: dendritic cells; IFN-γ: interferon γ; MCP1: monocyte chemoattractant protein 1; Th1/Th17: T helper 1/T helper 17 cells; TLR2: Toll-like receptor 2. For details, see text.

**Figure 2 biomedicines-13-00287-f002:**
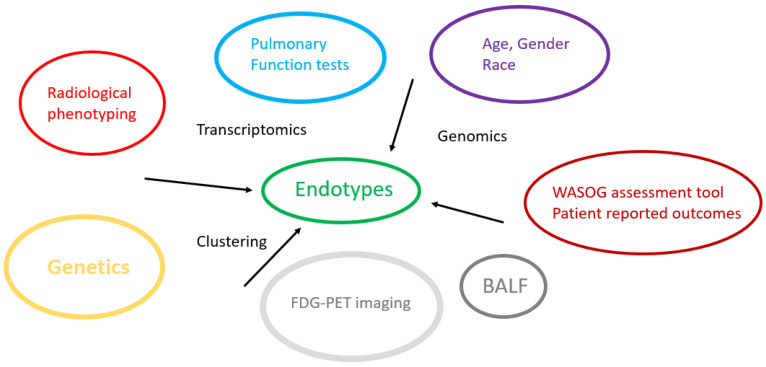
Endotyping and phenotyping in sarcoidosis.

**Table 1 biomedicines-13-00287-t001:** Risk factors of sarcoidosis.

*Genetic factors*
*HLA-DRB1*1101* [6]
*HLA-DRB1*03*, **0301* or **1501* (associated with Löfgren syndrome) [8]
*HLA-DRB1*07*, **14*, **15*, **01* and **03* and *DQB1*0602* (higher likelihood of progressive pulmonary sarcoidosis) [6]
TNF-α polymorphisms (*308AA* and *rs1800629*) [12,13,14,15]
Polymorphisms in Toll-like receptors (TLRs) [19,20]
*Butyrophilin-like-2* gene [22]
*Annexin A11* gene [24]
*Vimentin* gene [25]
*Infectious agents*
Mycobacterium species [26,28]
Propionibacterium (Cutibacterium) acnes [30,31]
*Occupational exposure*
Silica and metal dust [36,37]
Inhalation of organic bioaerosols (livestock, mold and industrial organic dusts) [33]
Insecticides [33]
*Lifestyle factors*
Obesity [38,51]
Diet [46]
*Seasonal exposure*
Increased incidence in winter and summer [49,50]

HLA: human leukocyte antigen; TNF-α: tumor necrosis factor-alpha.

**Table 2 biomedicines-13-00287-t002:** Complicated pulmonary sarcoidosis [143,144].

Parenchymal Disease	Airway Disease
Pulmonary fibrosis	Bullous emphysema
	Bronchiectasis
	Bronchial stenosis
Pulmonary Vascular Disease	Pleural Disease
Pulmonary hypertension	Pneumothorax
Pulmonary embolism	Pleural effusion
Coronary heart disease	Pleural thickening
Lung Cancer	Infections
	Aspergilloma
	Invasive aspergillosis
	Bacterial pneumonia
	COVID-19

**Table 3 biomedicines-13-00287-t003:** Different phenotype classifications in patients with sarcoidosis.

*Clinical parameters* Löfgren syndrome (erythema nodosum, bilateral hilar lymphadenopathy and acute-onset fever) [1,4] Heerfordt syndrome (anterior uveitis, bilateral parotid gland enlargement, facial nerve palsy and fever) [4] Clinical tools for phenotyping: Sarcoidosis Assessment Tool [147] WASOG organ assessment tool [148]	Pathognomonic of sarcoidosis. Young females, patients of Scandinavian origin and good prognosis Pathognomonic of sarcoidosis Likelihood of 15 organs involved in sarcoidosis (definite, probable and possible); histological confirmation of sarcoidosis required Update of the previous instrument and probability of organ involvement
*Ethnicity* Asian [101] Black [99]	Increased incidence of cardiac, muscular, renal and ocular manifestations Increased incidence of extra-pulmonary disease, progressive pulmonary disease, and liver and bone manifestations
*Radiological parameters* Scadding chest X-ray [134] HRCT phenotyping [143]	Five stages associated with resolution and prognosis: normal (0), bilateral hilar lymphadenopathy without pulmonary infiltrates (1), bilateral hilar lymphadenopathy with pulmonary infiltrates (2), pulmonary infiltrates without bilateral hilar lymphadenopathy (3) and extensive fibrosis with distortion or bullae (4). Non-fibrotic subtypes: multiple peri-bronchovascular, peri-fissural or subpleural micronodules; multiple larger peri-bronchovascular nodules; scattered larger nodules; consolidation as the predominant or sole abnormality Likely-to-be-fibrotic subtypes: bronchocentric reticulation with or without dense parenchymal opacification without cavitation, bronchocentric reticulation and dense parenchymal opacification with cavitation and large bronchocentric masses (i.e., PMF lookalike)
*Pulmonary function phenotyping* [145,146]	Normal, restrictive, obstructive or mixed pattern. Mixed disease associated with lower DLCO, stage 4 disease and higher mortality than seen in a purely obstructive defect; Restrictive is the most common pattern among Black individuals, while White individuals most commonly present normal PFTs; males frequently show obstruction and females restriction
*Nuclear imaging phenotyping* [154]	Four phenotypes:
	(1) Thoracic nodal hilar–mediastinal; (2) Thoracic nodal hilar–mediastinal and lungs; (3) Extended thoracic and extra-thoracic only nodal phenotype including inguinal–abdominal–supraclavicular stations; (4) All the above plus systemic organs and tissues such as muscles–bones–spleen and skin.
*Genetic phenotyping* [156]	Five phenotypes:
	(1) Abdominal organ involvement; (2) Ocular–cardiac–cutaneous–central nervous system disease involvement (OCCC); (3) Musculoskeletal–cutaneous involvement; (4) Pulmonary and intrathoracic lymph node involvement; (5) Extra-pulmonary involvement.

**Table 4 biomedicines-13-00287-t004:** Endotypes of sarcoidosis. For details, see text.

*HLADRB1*03*, **0301* or **1501* HLA polymorphism (*rs4143332*) [6,7,8,9,165]	Associated with Löfgren syndrome
*HLADRB1*07*, **14*, **15*, **01* and **03* and *DQB1*0602* [6,11]	Higher likelihood of progressive pulmonary sarcoidosis
*TNF-α* polymorphisms (*308AA and rs1800629*) [12,13,173]	Pulmonary disease progression and acute-onset disease
*ANXA11* locus *rs1049550* [169]	Protective effect by the minor T allele
Polymorphisms in TLRs (absence of the common haplotype in the *TLR10–TLR1–TLR6* gene cluster, *TLR3* L412F, *MyD88* and *CybB/Nox2*)	Chronic, persistent disease
*SEPP1* [158]	Worse lung function
*IL20RB*, *ABCC11*, *SFSWAP*, *AGBL4*, *miR-146a-3p* and *miR-378b* in a multi-omics model [158]	Associated with progressive sarcoidosis
*miR-21-5p*, *miR-340-5p* and *miR-212-3p* [160]	Differentiate patients with Löfgren syndrome
*miR-155*, *let-7c* and transcription factor *T-bet* [159]	Progressive disease
Th1, Th17, IFN-γ and NFAT signaling (*CD28*, *STAT1*, *CXCR3* and *CCR4* genes) [161]	Hilar lymphadenopathy
IL-2 and IL-7 pathways (*MRC2*, *SLC40A1*, *F2R*, *IL7*, *PTPN7*, *ADORA2A*, *SPRY2*, *PLA2G7* and *PTGS1* genes) [161]	More severe bronchial wall thickening
TGF-b1 and MTOR pathways (*TGFBR1*, *COL3A1*, *TLR3*, *ID1*, *TCF4*, *IGFBP6*, *PLA2G7*, *FADS1*, *ARGHAP12 and MMP10*, *SC5D*, *HIF1A* and *PPAR-α*) [161]	Parenchymal involvement—pulmonary fibrosis
Four gene modules [161]	4 novel endotypes: chronic sarcoidosis, hilar lymphadenopathy and acute lymphocytic inflammation, multi-organ involvement with increased immune response and extra-ocular involvement with PI3K activation
mTOR pathway activation [67,69,174]	Sarcoidosis progression
JAK/STAT pathway activation (17-gene signature) [71]	Sarcoidosis progression
Increased Th1 and Th17.1 cells [1,82]	Chronic sarcoidosis
Increased Th17 cells [80]	Löffler syndrome
High expression of CD25, CTLA4, CD69, PD-1 and CD95 in blood Tregs [89]	Chronic sarcoidosis

## Data Availability

No new data were created or analyzed in this study. Data sharing is not applicable to this article.

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
