# Peer review of "Phenotypes and Endotypes in Sarcoidosis: Unraveling Prognosis and Disease Course"

_biomedicines, 2025, doi:10.3390/biomedicines13020287_

Round 1
Reviewer 1 Report
Comments and Suggestions for Authors
Dear authors.
1. Since your review is very exclusive, please consider adding some lines on "dietary correlates" of sarcoidosis.
2. There reports that COVID-19 infection and vaccination impacted prognosis, occurrences and treatment of sarcoidosis. Please consider adding paragrpahs on current situation in term of covid-19 infection. See for instance, PMID: 39609037.
3. Please enrich your Table 1 with other risk factors and outcomes mentioned in 11PMID: 37439108.
Author Response
Dear authors.
- Since your review is very exclusive, please consider adding some lines on "dietary correlates" of sarcoidosis.
Thank you for your comment. We have added a paragraph regarding dietary correlates of sarcoidosis in section 2.4.
- There reports that COVID-19 infection and vaccination impacted prognosis, occurrences and treatment of sarcoidosis. Please consider adding paragrpahs on current situation in term of covid-19 infection. See for instance, PMID: 39609037.
Thank you very much for this comment. We have added an extra section regarding the correlation of sarcoidosis with COVID-19 (section 4.4). The suggested reference has also been added.
- Please enrich your Table 1 with other risk factors and outcomes mentioned in 11PMID: 37439108.
Thank you for your comment. We have added an extra table (table 2) on advanced sarcoidosis outcomes, including the impact of infections, as suggested in the Reference you suggested.
Reviewer 2 Report
Comments and Suggestions for Authors
The reviewed paper involves updating information of the phenotypes and endotypes of sarcoidosis, a rare multisystemic disease mainly affecting the lungs. The paper entails several aspects of the disease, including its etiological factors, basic and advanced diagnostic methods, as well as its molecular pathways. The following comments are suggested to improve the quality of the work before considering it for publication:
In the introductory part, authors have to elaborate on the definition of sarcoidosis, its prevalence, organ tropism, stages, and the importance of radiology in its staging.
An addition of a simple diagram summarizing all etiological factors is recommended.
In Table 1, delete the citations from the header of the table and place them inside the table next to each factor so that the reader can regain the source of the information more easily. The same applies to Table 2. In addition, the abbreviations in both tables should be listed in the footer section of each table.
The phenotyping and endotyping of the disease can be summarized in a diagram at the end of Section 6.
The conclusion should be updated with the future implications of the reviewed topic.
Author Response
We would like to thank the reviewer for the valuable comments. Below you can find our response to the comments.
The reviewed paper involves updating information of the phenotypes and endotypes of sarcoidosis, a rare multisystemic disease mainly affecting the lungs. The paper entails several aspects of the disease, including its etiological factors, basic and advanced diagnostic methods, as well as its molecular pathways. The following comments are suggested to improve the quality of the work before considering it for publication:
- In the introductory part, authors have to elaborate on the definition of sarcoidosis, its prevalence, organ tropism, stages, and the importance of radiology in its staging.
Thank you for your comment. We have expanded the introduction, presented the diagnosis of sarcoidosis and highlighted typical clinical and radiological findings. The prevalence and epidemiology are extensively described in section 2.6. The radiology staging is described in section 5.1, therefore we avoided to duplicate it in the introduction.
- An addition of a simple diagram summarizing all etiological factors is recommended.
Thank you for your comment. We have kept Table 1, as it is conclusive, instead of creating a new diagram.
- In Table 1, delete the citations from the header of the table and place them inside the table next to each factor so that the reader can regain the source of the information more easily. The same applies to Table 2. In addition, the abbreviations in both tables should be listed in the footer section of each table.
Thank you very much for your comment. The citations have been placed inside the tables and abbreviations are listed in the footer section.
- The phenotyping and endotyping of the disease can be summarized in a diagram at the end of Section 6.
Thank you for your valuable comment. We have added a figure (figure 2), summarizing phenotyping and endotyping of sarcoidosis
- The conclusion should be updated with the future implications of the reviewed topic.
Thank you for this important comment. We have updated the conclusion, suggesting that endotyping and phenotyping may allow the application of tailored treatments earlier in the disease course, minimizing the risk of toxicity.